# Active Seriation: Efficient Ordering Recovery with Statistical Guarantees

**James Cheshire**          **Yann Issartel**

LTCI, Télécom Paris, Institut Polytechnique de Paris

## Abstract

Active seriation aims at recovering an unknown ordering of $n$ items by adaptively querying pairwise similarities. The observations are noisy measurements of entries of an underlying $n \times n$ permuted Robinson matrix, whose permutation encodes the latent ordering. The framework allows the algorithm to start with partial information on the latent ordering, including seriation from scratch as a special case. We propose an active seriation algorithm that provably recovers the latent ordering with high probability. Under a uniform separation condition on the similarity matrix, optimal performance guarantees are established, both in terms of the probability of error and the number of observations required for successful recovery.

## 1 Introduction

The seriation problem involves ordering $n$ items based on noisy measurements of pairwise similarities. This reordering problem originates in archaeology, where it was used for the chronological dating of graves [Robinson, 1951]. More recently, it has found applications in data science across various domains, including envelope reduction for sparse matrices [Barnard et al., 1995], read alignment in de novo sequencing [Garriga et al., 2011, Recanati et al., 2017], time synchronization in distributed networks [Elson et al., 2004, Giridhar and Kumar, 2006], and interval graph identification [Fulkerson and Gross, 1965].

In many of these settings, pairwise measurements can be made in an adaptive fashion, leveraging information from previously chosen pairs. Motivated by these applications, we study the problem of recovering an accurate item ordering from a sequence of actively selected pairwise measurements.

The classical seriation problem has attracted substantial attention in the theoretical literature [Atkins et al., 1998, Fogel et al., 2013, Janssen and Smith, 2022, Giraud et al., 2023, Cai and Ma, 2023, Issartel et al., 2024]. In this line of work, the learner receives a single batch of observations, whereas in the active setting considered in this paper, the algorithm can adaptively select which pairs to observe; to the best of our knowledge, this active seriation setting has not been previously analyzed.

Related problems include adaptive ranking and sorting under noisy observations [Jamieson and Nowak, 2011, Braverman and Mossel, 2009, Heckel et al., 2019]. However, these problems typically rely on pairwise comparisons (e.g., is item $i$ preferred to item $j$?) to infer a total order. In contrast, seriation builds on pairwise similarity scores that encode proximity in the underlying ordering. This distinction leads to different statistical and algorithmic challenges.

**Problem setup.** In the seriation paradigm, we assume the existence of an unknown symmetric matrix $M$ representing pairwise similarities between a collection of $n$ items. The matrix $M$ is structured so that the similarities $M_{ij}$ are correlated with an unknown underlying ordering of the items, which is encoded by a permutation $\pi = (\pi_1, \ldots, \pi_n)$ of $[n]$. The similarity $M_{ij}$ between

39th Conference on Neural Information Processing Systems (NeurIPS 2025).

items $i$ and $j$ tends to be large when their positions $\pi_i$ and $\pi_j$ are close, and small when they are far apart. To model this structure formally, the literature assumes that $M$ is a Robinson matrix whose rows and columns have been permuted by the latent permutation $\pi$ [Fogel et al., 2013, Recanati et al., 2018, Janssen and Smith, 2022, Giraud et al., 2023]; see Section 2.1 for a formal presentation.

We consider a general framework in which the algorithm may be initialized with partial information on the latent ordering $\pi$. Specifically, it can be provided with a correct ordering of a subset of the items that is consistent with $\pi$. This setting includes seriation from scratch as the special case where no such information is given initially.

The algorithm adaptively selects pairs of items and observes noisy measurements of their similarities. A total of $T$ such observations are collected, and the noise is controlled by an unknown parameter $\sigma$. The goal is to recover the latent ordering $\pi$ of the $n$ items from these $T$ observations. We evaluate the performance of an estimator $\hat{\pi}$ by its probability of failing to identify $\pi$. See Section 2 for details.

**Contribution.** We introduce *Active Seriation by Iterative Insertion* (ASII), an active procedure for estimating the ordering $\pi$. Unlike most existing seriation methods, which are non-active, ASII is remarkably simple, runs in polynomial time, and yet achieves optimality guarantees both in terms of error probability and sample complexity.

In the general framework considered in this paper, we analyze the performance of ASII and establish exponential upper bounds on its probability of error. In the special case of seriation from scratch, where no prior ordering information is available, we provide a sharp characterization of the statistical difficulty of ordering recovery over the class of similarity matrices with minimal gap $\Delta$ between adjacent coefficients. This difficulty is governed by the signal-to-noise ratio (SNR) of order $\Delta^2 T/(\sigma^2 n)$, which can be interpreted as the number of observations per item, $T/n$, multiplied by the SNR per observation, $(\Delta/\sigma)^2$. Our results identify a phase transition at the critical level $\mathrm{SNR} \asymp \ln n$: below this threshold, ordering recovery is information-theoretically impossible, while above it, ASII achieves recovery with a probability of error that decays exponentially fast with the SNR. Moreover, we show that no algorithm can achieve a faster decay rate, establishing optimality in this regime.

Finally, we illustrate the performance of ASII through numerical experiments and a real-data application.

## 1.1 Related work

**Classical seriation.** The non-active seriation problem was first addressed by [Atkins et al., 1998] in the noiseless setting, using a spectral algorithm. Subsequent works analyzed this approach under noise [Fogel et al., 2013, Giraud et al., 2023, Natik and Smith, 2021], typically relying on strong spectral assumptions to establish statistical guarantees. More recent contributions proposed alternative polynomial-time algorithms with guarantees under different and sometimes weaker assumptions [Janssen and Smith, 2022, Giraud et al., 2023, Cai and Ma, 2023, Issartel et al., 2024]. Our analysis falls within the line of work on Lipschitz-type assumptions [Giraud et al., 2023, Issartel et al., 2024], as our $\Delta$-separation condition can be viewed as a lower Lipschitz requirement.

**Statistical-computational gaps.** Prior studies have suggested statistical–computational gaps in the non-active seriation problem [Giraud et al., 2023, Cai and Ma, 2023, Berenfeld et al., 2024], where known polynomial-time algorithms fall short of achieving statistically optimal rates under certain noise regimes or structural assumptions. While some of these gaps have recently been closed [Issartel et al., 2024], the resulting algorithms tend to be complex and may not scale well in practice. In contrast, in the active setting, we show that a simple and computationally efficient algorithm achieves statistically optimal performance.

**Bandit models.** In classical multi-armed bandits (MAB) [Bubeck and Cesa-Bianchi, 2012], each arm yields independent rewards, and the goal is to maximize cumulative reward or identify the best arm. In contrast, in active seriation, each query $(i, j)$ measures the similarity between two interdependent items, and all measurements must be consistent with a single latent ordering. This interdependence prevents a direct application of standard MAB algorithms such as UCB or Thompson Sampling, which treat arms as independent and do not exploit structural relationships between them.

Algorithmically, our approach is related to noisy binary search and thresholding bandits [Feige et al., 1994, Karp and Kleinberg, 2007, Ben-Or and Hassidim, 2008, Nowak, 2009, Emamjomeh-

Zadeh et al., 2016, Cheshire et al., 2020, Cheshire et al., 2021], which rely on adaptive querying under uncertainty. However, these methods operate on low-dimensional parametric models, whereas seriation involves a combinatorial ordering that must remain globally consistent across item pairs.

**Active ranking.** A related but distinct problem is active ranking [Heckel et al., 2019, Shah and Wainwright, 2017], where a learner infers a total order based on noisy pairwise comparisons or latent score estimates. Extensions include Borda, expert, and bipartite ranking [Saad et al., 2023, Cheshire et al., 2023]. These methods typically assume that each item is associated with an intrinsic scalar score, and that pairwise feedback expresses a directional preference between items. In contrast, seriation relies on pairwise similarity information, which encodes proximity rather than preference. Recovering the latent ordering therefore requires global consistency among all pairwise similarities, making the problem more constrained and structurally different from standard ranking tasks.

## 2 Active Seriation: Problem Setup

### 2.1 Similarity matrix and ordering

Given a collection of items $[n] := \{1, \ldots, n\}$, let $M = [M_{ij}]_{1 \leq i,j \leq n}$ denote the (unknown) *similarity matrix*, where the coefficient $M_{ij} \in \mathbb{R}$ measures the similarity between items $i$ and $j$. Our structural assumption on $M$ is related to the class of Robinson matrices, introduced below.

**Definition 2.1.** *A matrix $R \in \mathbb{R}^{n \times n}$ is called a Robinson matrix (or R-matrix) if it is symmetric and*

$$R_{i,j} > R_{i-1,j} \quad \text{and} \quad R_{i,j} > R_{i,j+1} ,$$

*for all $i \leq j$ on the upper triangle of $R$.*

The entries of a Robinson matrix decrease as one moves away from the (main) diagonal (see Figure 1, left). In other words, each row / column is unimodal and attains its maximum on the diagonal.

Following [Atkins et al., 1998], a matrix is said to be *pre-R* if it can be transformed into an R-matrix by simultaneously permuting its rows and columns (Figure 1, right). In this paper, we assume the similarity matrix $M$ is pre-R, i.e.,

$$M = R_\pi := [R_{\pi_i, \pi_j}]_{1 \leq i,j \leq n} , \qquad (1)$$

for some R-matrix $R$ and some permutation $\pi = (\pi_1, \ldots, \pi_n)$ of $[n]$. The permutation $\pi$ represents the latent *ordering* of the items.

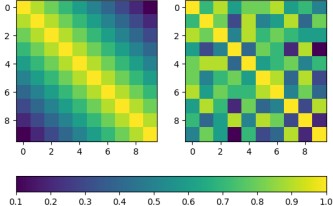

Figure 1: R-matrix & a permuted version.

In (1), the similarities $M_{ij}$ reflect the ordering $\pi$ as follows: $M_{ij}$ tends to be larger when the positions $\pi_i$ and $\pi_j$ are close together, and smaller when they are far apart.

**Remark 1.** *The items have exactly two orderings: if $\pi$ is an ordering, then the reverse permutation $\pi^{\mathrm{rev}}$, defined by $\pi_i^{\mathrm{rev}} = n + 1 - \pi_i$, is also an ordering. Indeed, if $M = R_\pi$, then $M = R_{\pi^{\mathrm{rev}}}^{\mathrm{rev}}$, where $R^{\mathrm{rev}}$ is obtained by reversing the rows and columns of $R$. In the sequel, we refer to either of these two orderings as the latent ordering.*

### 2.2 Active observation model

In the active seriation problem, the algorithm sequentially queries $T$ pairs of items and receives noisy observations of their pairwise similarities, encoded by the unknown similarity matrix $M$ in (1). The goal is to recover the latent ordering $\pi$ of the $n$ items from these noisy observations.

Initially, the algorithm is provided with partial information on the latent ordering $\pi$: it receives as input a correct ordering $\tilde{\pi}$ of the items $\{1, \ldots, n - \tilde{n}\}$, corresponding to the restriction of $\pi$ to this subset. This framework interpolates between online seriation ($\tilde{n} = 1$), where a single item is inserted into an existing ordering, and seriation from scratch ($\tilde{n} = n$).

More precisely, for some $\tilde{n} \in [n]$, the algorithm is given a permutation $\tilde{\pi} = (\tilde{\pi}_1, \ldots, \tilde{\pi}_{n-\tilde{n}})$ of $[n - \tilde{n}]$ satisfying[1]

$$\forall\, i, j \in [n - \tilde{n}]\,, \qquad \pi_i < \pi_j \quad \Longleftrightarrow \quad \tilde{\pi}_i < \tilde{\pi}_j\,. \tag{2}$$

In other words, the input permutation $\tilde{\pi}$ preserves the relative order of the items $\{1, \ldots, n - \tilde{n}\}$ in the latent ordering $\pi$.

At each round $t = 1, \ldots, T$, the algorithm selects a pair $(i_t, j_t)$ with $i_t \neq j_t$, possibly depending on the outcomes of previous queries. It then receives a noisy observation of the similarity $M_{i_t j_t}$, given by a $\sigma$-sub-Gaussian random variable[2] with mean $M_{i_t j_t}$. This sub-Gaussian setting covers standard observation models, including Gaussian noise and bounded random variables.

After $T$ queries, the algorithm outputs a permutation $\hat{\pi}$ of $[n]$ as its estimate of the latent ordering $\pi$.

## 2.3 Error probability and minimal gap

The algorithm is considered successful if $\hat{\pi}$ recovers either $\pi$ or its reverse $\pi^{\text{rev}}$, as both permutations are valid orderings (Remark 1). The probability of error is thus defined as:[3]

$$p_{M,T} := \mathbb{P}_{M,T}\left\{\hat{\pi} \neq \pi \text{ and } \hat{\pi} \neq \pi^{\text{rev}}\right\}\,, \tag{3}$$

where the probability is over the randomness in the $T$ observations collected on the matrix $M$.

A key quantity in our analysis of (3) is the separation between consecutive entries in the underlying R-matrix. Specifically, for any pre-R matrix $M$ as in (1), we define its *minimal gap* as

$$\Delta_M := \min_{1 < i < j \leq n-1} \left\{(R_{i,j} - R_{i-1,j}) \wedge (R_{i,j} - R_{i,j+1})\right\}\,, \tag{4}$$

where $\wedge$ denotes the minimum. The quantity (4) measures the smallest difference between neighboring entries in the R-matrix associated with $M$. Note that $\Delta_M > 0$ (by Definition 2.1) and that $\Delta_M$ is well-defined even though $M$ can be associated with different R-matrices (Remark 1).

Some of our guarantees are stated over classes of matrices with a prescribed minimal gap. Namely, for any $\Delta > 0$, we introduce

$$\mathcal{M}_\Delta := \left\{M \in \mathbb{R}^{n \times n} : M \text{ is pre-R, and } \Delta_M \geq \Delta\right\}\,, \tag{5}$$

as the set of pre-R matrices with minimal gap at least $\Delta$.

To simplify the presentation of our findings, we focus on the challenging regime where

$$\frac{\Delta}{\sigma} \leq 1\,, \qquad (\sigma > 0)\,, \tag{6}$$

i.e., the signal-to-noise ratio per observation is at most 1. This excludes mildly stochastic regimes where the problem has essentially the same difficulty as in the noiseless case.

## 3 Seriation procedure

ASII (Active Seriation by Iterative Insertion) reconstructs the underlying ordering by iteratively inserting each new item into an already ordered list. At iteration $k$, given a current estimated ordering $\hat{\pi}^{(k-1)}$ of the items $\{1, \ldots, k-1\}$, the algorithm inserts item $k$ at its correct position to form an updated ordering $\hat{\pi}^{(k)}$ of $\{1, \ldots, k\}$. This process is repeated until the full ordering of all $n$ items is obtained. The algorithm is fully data-driven: it takes as input only the sampling budget $T$; it does not rely on any knowledge of the noise parameter $\sigma$, the similarity matrix $M$, nor its minimal gap $\Delta_M$.

To perform this insertion efficiently, two key subroutines are used:

---

[1] In full generality, condition (2) should be assumed to hold either for the latent ordering $\pi$ or for its reverse $\pi^{\text{rev}}$; for simplicity of exposition, we state it in terms of $\pi$, as this distinction plays no essential role in the sequel.

[2] A random variable $X$ is said to be $\sigma$-*sub-Gaussian* if $\mathbb{E}[\exp(uX)] \leq \exp\left(\frac{u^2 \sigma^2}{2}\right)$ for all $u \in \mathbb{R}$.

[3] This lack of identifiability between $\pi$ and $\pi^{\text{rev}}$ persists in the seriation from scratch case, since condition (2) is vacuous.

(i) **Local comparison rule.** To decide where to insert $k$, the algorithm must compare its position relative to items already ordered in $\hat{\pi}^{(k-1)}$. This is achieved through the subroutine TEST, which determines whether $k$ lies to the left, in the middle, or to the right of two reference items $(l, r)$.

(ii) **Efficient insertion strategy.** To minimize the number of comparisons, the algorithm performs a binary search over the current ordering, where each comparison is made via TEST. Because these tests are noisy, the procedure is further stabilized through a backtracking mechanism.

**(i) Local comparison rule.** Given two items $(l, r)$ such that $\pi_l < \pi_r$, the goal is to determine whether $k$ lies to the left, in the middle, or to the right of $(l, r)$ in the unknown ordering $\pi$. Formally, this means deciding whether $\pi_k < \pi_l$, or $\pi_k \in (\pi_l, \pi_r)$, or $\pi_k > \pi_r$, respectively.

The subroutine TEST is based on the following property of Robinson matrices: when $k$ lies between $l$ and $r$, its similarity to both $l$ and $r$ tends to be higher than the similarity between $l$ and $r$ themselves. Accordingly, TEST compares the three empirical similarities $\widehat{M}_{kl}$, $\widehat{M}_{kr}$, and $\widehat{M}_{lr}$, and identifies the smallest one as the pair of items that are farthest apart. For example, if $\widehat{M}_{lr}$ is the smallest, then $k$ lies in the middle. The pseudocode of TEST is provided in Appendix B.

Each call to the subroutine TEST is performed with a limited sampling budget, typically of order $O(T/(\tilde{n} \log_2 k))$, in order to be sampling-efficient. With such a budget, individual tests are not designed to be reliable with high probability. Higher reliability is required only at a few critical steps of ASII (e.g., during the initialization of the binary search), where a larger budget of order $O(T/\tilde{n})$ is allocated to ensure correctness with high probability. This design trades local test accuracy for global budget efficiency; occasional incorrect tests are later corrected by a backtracking mechanism.

**(ii) Efficient insertion strategy.** The idea of incorporating backtracking into a noisy search has appeared in several studies, e.g., [Feige et al., 1994, Ben-Or and Hassidim, 2008, Emamjomeh-Zadeh et al., 2016, Cheshire et al., 2021]. Here, we adopt this general principle to design a robust insertion mechanism that remains reliable under noisy relational feedback.

At this stage of the algorithm, the items $\{1, \ldots, k-1\}$ have already been placed into an estimated ordering $\hat{\pi}^{(k-1)} = (\hat{\pi}_1^{(k-1)}, \ldots, \hat{\pi}_{k-1}^{(k-1)})$ constructed so far. To insert the new item $k$ into this list, we use the subroutine BINARY & BACKTRACKING SEARCH (BBS), which determines the relative position of $k$ within the current ordering $\hat{\pi}^{(k-1)}$. The search proceeds by repeatedly using the subroutine TEST to decide whether $k$ lies in the left or right half of a candidate interval, thereby narrowing down the possible insertion range.

Because the outcomes of TEST are noisy, even a single incorrect decision can misguide the search and lead to an erroneous final placement. A naive fix would be to allocate many samples per TEST to ensure highly reliable outcomes, but this would increase the sample complexity and undermine the benefit of active sampling. Instead, BBS uses a small number of samples per TEST, of order $O(T/(\tilde{n} \log_2 k))$, just enough to ensure a constant success probability (e.g., around $3/4$). Backtracking then acts as a corrective mechanism that prevents local errors from propagating irreversibly.

The backtracking mechanism operates as follows: the algorithm keeps track of previously explored intervals and performs sanity checks at each step to detect inconsistencies in the search path. When an inconsistency is detected, it backtracks to an earlier interval and resumes the search. This prevents local mistakes from propagating irreversibly. Theoretical analysis shows that, as long as the number of correct local decisions outweighs the number of incorrect ones — an event that occurs with high probability under the assumption $M \in \mathcal{M}_\Delta$ — the final insertion position is accurate.

Hence, the backtracking mechanism allows the algorithm to detect and correct occasional local errors. As a result, we can use only a small number of samples per call to TEST, while still ensuring correct insertions at the global level. The ASII procedure thus provides both sampling efficiency and robustness despite noisy observations. A pseudocode of this procedure is given in Appendix B.

## 4   Performance analysis

We study the fundamental limits of ordering recovery in active seriation, deriving information-theoretic lower bounds and algorithmic upper bounds on the error probability defined in (3).

## 4.1 Upper bounds for ASII

We analyze the performance of ASII when the algorithm is provided with a permutation $\tilde{\pi}$ of the items $\{1, \ldots, n - \tilde{n}\}$, consistent with the latent ordering $\pi$ of the $n$ items. The following theorem gives an upper bound on the error probability of ASII for recovering $\pi$. Its proof is in Appendix C.

**Theorem 4.1** (Upper bound with partial information). *There exists an absolute constant $c_0 > 0$ such that the following holds. Let $n \geq 3$ and $\tilde{n} \in [n]$, and assume that the input permutation $\tilde{\pi}$ of $[n - \tilde{n}]$ satisfies* (2). *Let $(\Delta, \sigma, T)$ be such that condition* (6) *holds and*

$$\frac{\Delta^2 T}{\sigma^2 \tilde{n}} \geq c_0 \ln n \ .$$

*If $M \in \mathcal{M}_\Delta$, then the error probability of* ASII *satisfies*

$$p_{M,T} \leq \exp\left( - \tfrac{1}{800} \frac{\Delta^2 T}{\sigma^2 \tilde{n}} \right) \ . \tag{7}$$

Once the SNR, $\Delta^2 T / (\sigma^2 \tilde{n})$, exceeds a logarithmic threshold in $n$, the error probability of ASII decays exponentially fast with the SNR. We emphasize that ASII achieves this performance without requiring any knowledge of the model parameters $(\Delta, \sigma)$.

Beyond this uniform guarantee over the class $\mathcal{M}_\Delta$, the performance of ASII admits a finer, instance-dependent characterization. The same bound holds with $\Delta$ replaced by the true minimal gap $\Delta_M$ of the underlying matrix $M$ (defined in (4)). Precisely, for the instance-dependent $\mathrm{SNR}_M = \Delta_M^2 T / (\sigma^2 \tilde{n})$, the bound becomes $p_{M,T} \leq \exp(-\mathrm{SNR}_M / 800)$ whenever $\mathrm{SNR}_M \geq c_0 \ln n$.

## 4.2 Minimax optimal rates for seriation from scratch

We establish matching information-theoretic lower bounds in the case of active seriation from scratch ($\tilde{n} = n$), where no prior ordering information is available. This analysis identifies two regimes, depending on whether the SNR is below or above a critical threshold.

### 4.2.1 Impossibility regime.

When the SNR satisfies $\frac{\Delta^2 T}{\sigma^2 n} \lesssim \ln n$, no algorithm can recover the ordering with vanishing error probability. The following theorem formalizes this impossibility, establishing a constant lower bound on the error probability (3) for any algorithm in this regime.

**Theorem 4.2** (Impossibility regime). *There exists an absolute constant $c_1 > 0$ such that the following holds. Let $n \geq 9$ and $(\Delta, \sigma, T)$ be such that*

$$\frac{\Delta^2 T}{\sigma^2 n} \leq c_1 \ln n \ .$$

*Then, for any algorithm $A$, there exists a matrix $M \in \mathcal{M}_\Delta$ such that the error probability of $A$ satisfies $p_{M,T} \geq 1/2$.*

As expected, the impossibility regime is more pronounced when the minimal gap $\Delta$ is small or when the noise parameter $\sigma$ is large. To escape this regime, the number of observations per item, $T/n$, must grow at least quadratically with $\sigma / \Delta$ (up to logarithmic factors). The proof is in Appendix F.

### 4.2.2 Recovery regime.

In the complementary regime where the SNR satisfies $\frac{\Delta^2 T}{\sigma^2 n} \gtrsim \ln n$, exact recovery becomes achievable. The ASII algorithm attains an exponentially small error probability, and this rate is minimax optimal over the class $\mathcal{M}_\Delta$, up to absolute constants in the exponent.

Specifically, the upper bound in this regime follows directly from Theorem 4.1 by taking $\tilde{n} = n$. If

$$\frac{\Delta^2 T}{\sigma^2 n} \geq c_0 \ln n \ , \tag{8}$$

where $c_0$ is the same absolute constant as in Theorem 4.1, then for any $M \in \mathcal{M}_\Delta$,

$$p_{M,T} \leq \exp\left( - \tfrac{1}{800} \frac{\Delta^2 T}{\sigma^2 n} \right) \ .$$

Conversely, the next theorem gives a matching lower bound, showing that no algorithm can achieve a faster error decay than exponential in the SNR. Its proof is in Appendix F.

**Theorem 4.3** (Recovery regime). *Let $n \geq 4$ and $(\Delta, \sigma, T)$ be such that condition (8) holds. Then, for any algorithm $A$, there exists $M \in \mathcal{M}_\Delta$ such that the error probability of $A$ satisfies*

$$p_{M,T} \geq \exp\left(-8\frac{\Delta^2 T}{\sigma^2 n}\right) .$$

Together, Theorem 4.2 and 4.3 delineate the statistical landscape of active seriation from scratch, establishing a sharp phase transition between impossibility and recovery at the critical SNR level $\frac{\Delta^2 T}{\sigma^2 n} \asymp \ln n$.

**4.2.3 Discussion: intrinsic hardness and invariance to model assumptions.** Both lower bounds (Theorems 4.2 and 4.3) are established under a Gaussian noise model with centered, homoscedastic entries of variance $\sigma^2$, whereas our upper bound is proved in the more general sub-Gaussian setting allowing heterogeneous noise levels. Since these bounds match, potential heterogeneity in the noise variances does not affect the minimax rates (at least in terms of exponential decay in SNR).

Moreover, the lower bounds are derived for the simple, affine, Toeplitz matrix

$$R_{ij} = (n - |i - j|)\Delta , \tag{9}$$

yet the attainable rates coincide with those obtained under the general assumption $M \in \mathcal{M}_\Delta$. Hence, allowing heterogeneous, non-Toeplitz matrices comes at no statistical cost. This may appear surprising, since the Toeplitz assumption is classical in batch seriation (e.g., [Cai and Ma, 2023]).

Even when the latent matrix is fully known, as in the one-parameter family (9) with known parameter $\Delta$, the attainable rates remain unchanged. This indicates that the hardness of active seriation arises from the combinatorial nature of the latent ordering, rather than from uncertainty about the latent matrix.

## 4.3 Sample complexity for high probability recovery

We summarize our results in terms of sample complexity, defined as the number of observations required to achieve exact recovery with probability at least $1 - 1/n^2$. Combining the impossibility result of Theorem 4.2 with the recovery guarantee of Theorem 4.1, we obtain a sharp characterization of the sample complexity in the from-scratch case ($\tilde{n} = n$).

**Corollary 4.4** (Sample complexity). *Let $n \geq 3$ and $\tilde{n} \in [n]$, and assume that the input permutation $\tilde{\pi}$ of $[n - \tilde{n}]$ satisfies (2). Then ASII achieves exact recovery with probability at least $1 - 1/n^2$, if*

$$T \gtrsim \frac{\sigma^2 \tilde{n} \ln n}{\Delta^2} .$$

*In particular, in the from-scratch case $\tilde{n} = n$, the minimax-optimal number of observations required for exact recovery with probability at least $1 - 1/n^2$, satisfies $T^\star \asymp \frac{\sigma^2 n \ln n}{\Delta^2}$.*

Thus, in active seriation from scratch, ASII attains the minimax-optimal sample complexity $T^\star$, which depends transparently on the problem parameters $(\Delta, \sigma, n)$. Crucially, ASII can achieve high probability recovery with a number of queries $T \ll n^2$, highlighting a substantial advantage over the classical batch setting where all $n^2$ pairwise similarities are observed.

## 4.4 Extension beyond uniform separation

Our analysis so far has focused on exact recovery under the uniform separation assumption $M \in \mathcal{M}_\Delta$. While this assumption is natural for characterizing the fundamental limits of exact recovery, it is also idealized: in practice, some items may be nearly indistinguishable in terms of their pairwise similarities, making their relative ordering statistically impossible to recover.

We therefore consider arbitrary pre-R matrices $M$, as defined in model (1), without any separation assumption. Even in this setting, a direct extension of the ASII procedure continues to provide meaningful and statistically optimal guarantees: it correctly recovers the relative ordering of items that are sufficiently well separated.

**Algorithmic extension of ASII.** The procedure follows the same iterative insertion scheme as our ASII algorithm, but takes as input a tolerance parameter $\tilde{\Delta} > 0$, which sets the resolution at which the algorithm attempts to distinguish items. Now, when ASII identifies a candidate insertion location for a new item in the current ordering, this location is checked through an additional high-probability validation test at precision $\tilde{\Delta}/2$. The item is inserted if the test succeeds; otherwise it is discarded and does not appear in the final output.

As a consequence, this extension no longer outputs a permutation $\hat{\pi}$ of all $n$ items. Instead, it returns a subset $S \subset [n]$ together with a permutation $\hat{\pi}_S$ of $S$, which serves as an estimator of the relative ordering of the items in $S$. The corresponding pseudo-code is deferred to the appendix.

**Robustness beyond uniform separation.** We now state robustness guarantees for the extension described above. To this end, we introduce a notion of $\Delta$-maximality, inspired by the class $\mathcal{M}_\Delta$ defined in (5). Specifically, for any subset $S \subset [n]$, we write $M_S$ for the submatrix of the similarity matrix $M$ restricted to the items in $S$. By a slight abuse of notation, we write $M_S \in \mathcal{M}_\Delta$ to mean that $M_S$ is pre-R and satisfies $\Delta_{M_S} \geq \Delta$, where $\Delta_{M_S}$ is the minimal gap defined in (4).

**Definition 4.5** ($\Delta$-maximal subset). *For any $\Delta > 0$, a subset $S \subset [n]$ is said to be $\Delta$-maximal if $M_S \in \mathcal{M}_\Delta$ and $M_{S \cup \{k\}} \notin \mathcal{M}_\Delta$ for all $k \in [n] \setminus S$.*

Intuitively, a $\Delta$-maximal subset cannot be enlarged without adding items that are too similar (within $\Delta$) to those already in $S$.

We now show that the guarantee of Theorem 4.1 can be extended beyond the uniformly separated setting, to any pre-R matrix $M$, by operating at a user-chosen resolution level $\tilde{\Delta}$. The conditions on the parameters are the same as in Theorem 4.1, with $\Delta$ replaced by the input parameter $\tilde{\Delta}$.

**Theorem 4.6** (Guarantees beyond uniform separation). *There exist absolute constants $c, c' > 0$ such that the following holds. Let $n \geq 3$, $\tilde{n} \in [n]$ and the input permutation $\tilde{\pi}$ of $[n - \tilde{n}]$ satisfy (2). Let $(\tilde{\Delta}, \sigma, T)$ be such that $\tilde{\Delta}/\sigma \leq 1$ and $\frac{\tilde{\Delta}^2 T}{\sigma^2 \tilde{n}} \geq c \ln n$. Then, for any pre-R matrix $M \in \mathbb{R}^{n \times n}$, the extension of ASII outputs, with probability at least $1 - \exp\left(-c' \frac{\tilde{\Delta}^2 T}{\sigma^2 \tilde{n}}\right)$, a $\tilde{\Delta}$-maximal subset $S \subset [n]$ in the sense of Definition 4.5, and a correct ordering $\hat{\pi}_S$ of the items in $S$.*

## 5 Empirical results

We illustrate the behavior of ASII through numerical experiments and a real-data example.

**Numerical simulations.** We assess the empirical behavior of ASII on synthetic data and compare it to three benchmark methods: (i) the batch seriation algorithm ADAPTIVE SAMPLING [Cai and Ma, 2023], (ii) the batch SPECTRAL SERIATION [Atkins et al., 1998], and (iii) an active, naive insertion variant of ASII without backtracking. Since, to the best of our knowledge, the literature does not contain seriation methods designed for the active setting, these three procedures serve as reference points. All methods are evaluated under identical sampling budgets on four representative scenarios, covering both homogeneous (Toeplitz) and non-homogeneous (non-Toeplitz), Robinson matrices. Full experimental details are deferred to Appendix H.

Figure 2 provides a visual illustration of the four scenarios considered. Scenario (1) corresponds to a Toeplitz Robinson structure, while the remaining scenarios (2-3-4) depart from the Toeplitz assumption and exhibit more heterogeneous Robinson geometries.

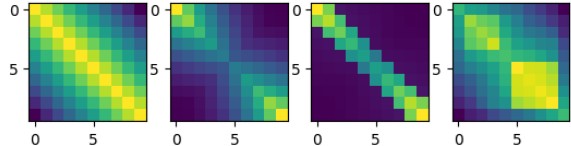

Figure 2: Robinson matrices for scenarios (1)-(4), from left to right.

Figure 3 reports the empirical probability of error of all methods as a function of the minimal gap $\Delta$. Across all four scenarios, ASII consistently outperforms the naive iterative insertion procedure,

highlighting the benefits of its backtracking corrections; these gains are consistent with the logarithmi improvement predicted by our theoretical analysis.

As expected, in scenario (1), where the underlying matrix is Toeplitz, ASII performs below the two batch methods, which are known to perform well in this setting. This behavior can be attributed to the fact that ASII is designed for general Robinson matrices and does not exploit Toeplitz regularity (unlike its competitors); moreover, its sampling budget is distributed across multiple binary-search iterations, which can be less efficient than batch sampling. This does not contradict our theoretical guarantees, which establish rates up to absolute constants that are not characterized by the analysis and may be large for active, iterative procedures such as ASII.

In contrast, in the heterogeneous (non-Toeplitz) scenarios (2-3-4), ASII remains consistently accurate, whereas both batch methods exhibit unstable behavior and may fail entirely in some cases.

Overall, these experiments illustrate that ASII maintains stable empirical performance across various scenarios, and can be particularly effective on matrices with localized variations (scenarios 2-3-4), where batch methods tend to struggle.

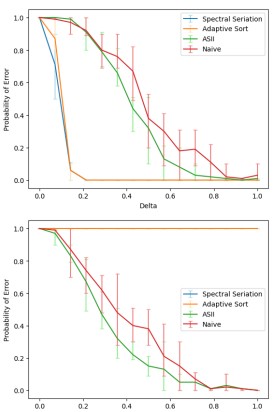
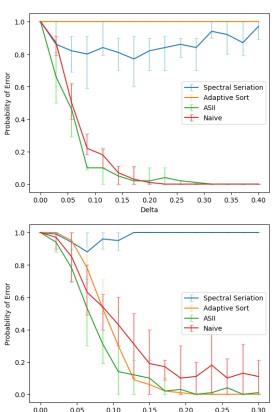

Figure 3: Empirical error probabilities for ASII and three benchmark methods as the parameter $\Delta$ varies. Scenarios (1-2-3-4) are displayed from left to right and top to bottom. Each experiment uses $n = 10$ items and $T = 10,000$ observations. For each value of $\Delta$, 100 Monte Carlo runs are split into 10 equal groups; error bars show the 0.1 and 0.9 quantiles of the empirical error across groups.

**Application to real data.** We further assess the robustness of ASII on real single-cell RNA sequencing data (human primordial germ cells, from [Guo et al., 2015], previously analyzed by [Cai and Ma, 2023]). Although such biological data depart substantially from the idealized Robinson models assumed in our theory, ASII still produces a meaningful reordering of the empirical similarity matrix, revealing coherent developmental trajectories among cells. This example highlights the potential practical relevance of the proposed approach beyond the stylized assumptions of our theoretical framework. Full experimental details are provided in Appendix H.

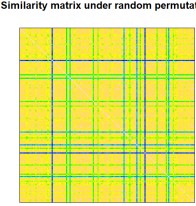
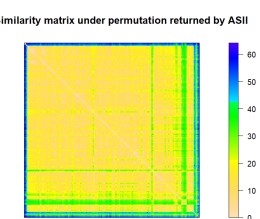

Figure 4: Similarity matrix of a single-cell RNA-seq dataset before and after reordering by ASII.

# 6 Discussion

This work introduces an active-learning formulation of the seriation problem, together with sharp theoretical guarantees and a simple polynomial-time algorithm. We characterize a phase transition in sample complexity governed by the $\mathrm{SNR} = \Delta^2 T/(\sigma^2 n)$: recovery is impossible when $\mathrm{SNR} \lesssim \ln n$, while ASII achieves near optimal performances once this threshold is exceeded. Our analysis highlights how adaptive sampling combined with corrective backtracking can substantially improve statistical efficiency. We now turn to a discussion of several complementary aspects of the problem.

**Noise regimes.** Our analysis focused on the stochastic regime where the per-observation signal-to-noise ratio $\Delta/\sigma$ is at most 1, which captures the most challenging setting for active seriation. The results, however, extend naturally to less noisy regimes ($\Delta/\sigma > 1$): in that case, accurate recovery requires only $T \gtrsim n \ln n$ queries, reflecting the intrinsic $O(n \ln n)$ cost of performing $n$ adaptive binary insertions. Further details are provided in Appendix A.1.

**Gain from active learning.** Our active framework enables recovery of the underlying ordering without observing the entire similarity matrix. Whereas batch approaches require $O(n^2)$ observations, our active algorithm ASII succeeds with only $T \gtrsim (\sigma/\Delta)^2 n \ln n$ samples. This corresponds to a fraction $\frac{\ln n}{(\Delta/\sigma)^2 n}$ of the full matrix and yields a substantial reduction in sample complexity, provided that $\Delta/\sigma$ remains bounded away from zero. This gain arises from the ability of adaptive sampling to draw information from a well-chosen, small subset of pairwise similarities, from which the entire matrix can be reordered, achieving strong statistical guarantees under limited sampling budgets.

In certain scenarios, when differences between candidate orderings are highly localized, our active algorithm can succeed under weaker signal conditions than some batch methods, see Appendix A.2 for a detailed comparative example.

**Fixed-budget formulations.** Throughout this work, we focused on the fixed-budget setting, where the total number of samples $T$ is fixed in advance and the objective is to minimize the error probability within this budget. This way, the algorithm does not require prior knowledge of the noise parameter $\sigma$, nor of the minimal signal gap $\Delta_M$ of the latent matrix $M$; it simply allocates the available budget $T$ across tests. Yet, its performance depends on the unknown $\sigma$ and $\Delta_M$ through the signal-to-noise ratio $\mathrm{SNR}_M = \Delta_M^2 T/(\sigma^2 n)$, which determines the achievable accuracy.

**Potential applications.** Seriation techniques are broadly relevant in domains where pairwise similarity information reflects a latent one-dimensional structure. Examples include genomic sequence alignment, where seriation helps reorder genetic fragments by similarity, and recommendation systems, where item-item similarity matrices can reveal latent preference orderings. We also illustrated, on real single-cell RNA sequencing data, that ASII can recover biologically meaningful trajectories despite the data departing strongly from our theoretical model. These settings often involve noisy or costly pairwise measurements, for which active seriation provides an appealing alternative to batch reordering methods.

**Future directions.** We also proposed an extension to settings where the uniform separation assumption does not hold, in which ASII outputs an ordering of a subset of well-separated items. How to exploit more global ordering structure beyond this setting is an important open question.

Another natural extension is to study the fixed-confidence setting, where the algorithm must adaptively decide when to stop sampling in order to achieve a prescribed confidence level. Such a formulation would typically require variance-aware sampling policies and data-driven stopping rules, possibly involving online estimation of $\sigma$. Developing such an adaptive, fixed-confidence version of ASII is an interesting avenue for future work.

### Acknowledgements

The work of J. Cheshire is supported by the FMJH, ANR-22-EXES-0013.

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
