# OpenReview forum: "Active Seriation: Efficient Ordering Recovery with Statistical Guarantees"
_NeurIPS.cc/2025/Conference — NeurIPS 2025 poster_

### Official Review · Reviewer_1ga3 · 2025-06-26

**Clarity:** 3
**Significance:** 3
**Originality:** 3
**Rating:** 4
**Confidence:** 2

**Summary:**

The paper addresses the problem of recovering the correct order of items using noisy pairwise similarity comparisons, where the comparisons are chosen actively. It introduces a simple and efficient algorithm called ASII (Active Seriation by Iterative Insertion), which incrementally builds the ordering using binary search with backtracking. The authors provide theoretical guarantees showing that ASII achieves optimal recovery rates under certain signal-to-noise conditions, and they validate its performance through simulations.

**Questions:**

1.	Could the authors clarify whether the algorithm assumes sigma is known, and if not, how T_0 is chosen in practice? If sigma is unknown, can the algorithm be made adaptive to it?
2.	The algorithm’s performance degrades when the minimal gap (delta) is small or varies across the matrix, while batch methods like Adaptive Sorting are more robust in such cases. Can the authors comment on whether ASII could be modified to incorporate more global or aggregate information to improve robustness in heterogeneous matrices? Even a heuristic or hybrid approach that combines local and global signals could be valuable.
3.	Could ASII be extended to handle partial or missing similarity data?

**Ethical Concerns:**

["NO or VERY MINOR ethics concerns only"]

**Final Justification:**

I have reviewed the authors' response, added additional questions, and also read the other reviews. I have decided to maintain my score.

**Limitations:**

Yes

**Quality:**

3

**Strengths And Weaknesses:**

Strengths –
1.	The paper introduces and formalizes the problem of active seriation, which is a natural and important extension of classical seriation to settings where pairwise similarities can be queried adaptively.
2.	The authors provide clear and tight theoretical guarantees. They show exactly when recovery is possible and when it’s not, and their algorithm matches these limits up to constants.
3.	The link to multi-armed bandits, especially thresholding bandits, is clever and helps ground the analysis.
4.	The paper does a good job of providing intuitive explanations alongside the formal results. For example, the discussion around how the minimal gap (delta) and the noise level (sigma) affect the ability to recover the true ordering helps the reader build a solid understanding of the underlying difficulty of the problem. These kinds of insights make the theoretical contributions more accessible and meaningful.
5.	The supplementary material includes detailed and well-organized proofs. It’s clear the authors have done a careful job with the analysis.
6.	The paper is well-written and easy to follow. Definitions, algorithms, and results are all clearly presented.



Weaknesses –
1.	All experiments are conducted on synthetic data. The absence of real-world datasets makes it difficult to assess the practical utility and robustness of the method.
2.	While the paper compares ASII to Adaptive Sorting and Spectral Seriation, it does not benchmark against more recent or diverse active learning or ranking algorithms.
3.	The algorithm and its guarantees (look like) assume knowledge of the noise variance (sigma), which may not be available or easy to estimate in practice. In the ‘Test’ subroutine, the number of samples T_0 used per test is chosen to ensure high-probability correctness, which should depend on the noise level (sigma).
4.	While the algorithm is adaptive to the unknown minimal gap (delta), it does not adapt to unknown noise levels. This limits its applicability in settings where sigma is not known or cannot be estimated reliably.
5.	While ASII performs well when the minimal gap (delta) is reasonably large and uniform, its reliance on local comparisons makes it less effective when (delta) is very small or varies significantly across the matrix. In such cases, batch algorithms like Adaptive Sorting, which leverage aggregate information across the entire matrix, can be more robust. This is evident in the experimental results (e.g., Scenario 2), where ASII’s performance degrades more sharply than that of its batch counterparts.

---

> ### Author Rebuttal · Authors · 2025-07-31
>
> Thank you for your thoughtful feedback, and for taking the time to review our work.
>
> ---
>
> #  Response to Reviewer's Comment on Noise Variance $\sigma$ (Weaknesses 3 and 4, Question 1)
>
> Our algorithm does *not* assume prior knowledge of the noise variance
>  $\sigma$.
> Instead, the sample size
> $T_0$
>  used in the **Test** subroutine is determined adaptively based on the total sample budget
> $T$,
> which is allocated as
> 1. $T_0 = \left\lfloor \frac{T}{3n} \right\rfloor$ for the initial tests, and
> 2. $T_0 = \left\lfloor \frac{T}{3n T_k} \right\rfloor = \left\lfloor \frac{T}{9 n \log k} \right\rfloor$ for tests within the **Binary \& Backtracking Search** subroutine.
>
> This approach does not require explicit knowledge of
> $\sigma$. Instead, the total budget
>  $T$
>  should be large enough (e.g., satisfying $T \gtrsim \frac{\Delta^2 n \log n}{\sigma^2}$ as in Theorem 3.2(a)) to ensure that each test in 2 (above)  obtains a sufficiently accurate estimate with constant probability of success (better than random guessing).
>
> An exception is the initial test 1 (above), which decides whether to invoke the **Binary \& Backtracking Search** subroutine. Since this decision is not revisited, the test must be sufficiently reliable to avoid incorrect branching. This initial test outcome will be correct with high probability once the total budget satisfies $T \gtrsim \frac{\Delta^2 n \ln n}{\sigma^2}$  as in Theorem 3.2(a).
>
> In practice, this means that
> $T_0$
>  scales with the total budget
> $T$
>  and number of items
>  $n$,
> allowing the algorithm to adapt implicitly to the noise level without needing
> $\sigma$
> as an input parameter.
>
> We acknowledge that this point was not sufficiently emphasized in the submitted version of the paper. We will revise the manuscript accordingly .
>
> ---
>
>
>
> #  Response to Reviewer’s Comment on Dependence on Minimal Gap $\Delta$ (Weakness 5, Question 2)
>
> We agree that robustness to variations in the minimal gap $\Delta$ is an important issue. The current **ASII** algorithm relies heavily on local measurements, such as pairwise comparisons between items. Specifically, to find the relative position of item $k$ with respect to a pair $(l,r)$ of reference items, the **Test** subroutine compares the corresponding pairwise similarities $M_{k,l}, M_{k,r}, M_{l, r}$. This approach can be sensitive to variations in the minimal gap, particularly when $\Delta$ is small. As a result, when $\Delta$ decays, the algorithm may struggle to maintain stable performance.
>
> To improve robustness, one potential solution is to modify the **Test** subroutine to incorporate more global information. For instance, instead of just comparing pairwise associations ($M_{k,l}, M_{k,r}, M_{l, r}$), we could compute global distances such as the $L_1$ distance between items. Specifically, we could define the $L_1$ distance between items $k$ and $l$ as the difference between their respective rows in the matrix, i.e., $D_{k,l} = \lVert M_{k} - M_{l} \rVert_1$. By using these distances, we could compare $D_{k,l}, D_{k,r}, D_{l,r}$ rather than directly comparing the pairwise associations ($M_{k,l}, M_{k,r}, M_{l, r}$). This hybrid approach, inspired by the batch algorithm of Cai and Ma (2022), could offer more robust performance in heterogeneous matrices.
>
> However, since our method is active and aimed at reducing sample complexity, we must balance the incorporation of global measurements with the need to minimize the number of samples. To achieve this, we propose estimating global measurements via random subsampling, using a small number of entries (e.g., $\log(n)$ samples) to estimate the $L_1$ distance. This would allow us to incorporate global signals while keeping the computational cost low compared to batch methods
>
> We plan to include numerical experiments with this more robust method in the revised version of the paper and, ideally, to provide theoretical results on its performance.
>
>
> ---
>
> # Response to Reviewer’s Comment on Missing Data (Question 3)
>
> Yes, it is possible to extend the **ASII** algorithm to handle partial or missing similarity data. As mentioned in our response to the previous question, one approach is to modify the **Test** subroutine to incorporate more global measurements, such as the $L_1$ distance between items, using random subsampling. This technique could  be adapted to handle missing data. By drawing a subsample of data points (e.g., $\log(n)$ entries), we can estimate  distances, even when some values are missing.
>
> To ensure robustness in the presence of missing data, we would need to make certain assumptions about the proportion of missing entries. Specifically, as long as the proportion of missing data is small, we can select a sufficiently large subsample to ensure that the remaining valid data points are representative. This approach would allow the algorithm to continue making decisions with a reasonably high probability of success, while mitigating the effects of missing data.
>
> We plan to explore this extension in our numerical experiments and, if possible, include a theoretical analysis of this method.
>
> ---
>
> # Response to Reviewer’s Comment on Real-World Applications (Weaknesses 1)
>
> We appreciate the reviewer's suggestion to highlight real-world applications of our method. To strengthen the significance of our work, we refer to the recent study by Cai and Ma (2022), which applies seriation reordering techniques to real-world biological data. In their study, two datasets were analyzed to infer the temporal ordering of cells in single-cell RNA sequencing experiments. We plan to apply our active seriation algorithm to these same datasets and present our results in the updated version of our paper, comparing the performance with that of Cai and Ma’s method.
>
> 1. **Real Dataset 1:** The first dataset consists of 372 primary human skeletal muscle myoblasts undergoing differentiation at 24-hour intervals. The goal was to infer the temporal order of cells during differentiation. By constructing similarity matrices and applying a seriation algorithm,  Cai and Ma (2022) demonstrated the good performance of their algorithm in recovering the true temporal order.
>
> 2. **Real Dataset 2:** The second dataset involves 149 human primordial germ cells from various ages ranging from 4 to 19 weeks. A similar methodology was used to infer the temporal order of these cells.
>
>
> These real-world applications highlight the practical utility of seriation algorithms in analyzing complex, noisy, and high-dimensional biological data. The ability to recover latent temporal or sequential structures from such datasets is a crucial problem in computational biology. Our method offers a computationally efficient and statistically robust solution, particularly when data is  noisy. Moreover, one of the key advantages of our active seriation algorithm is its ability to achieve accurate results with fewer similarity measurements, which can be of practical interest in scenarios where obtaining large amounts of pairwise comparisons may be costly or time-consuming.
>
> Additionally, we highlight several other domains where our seriation method can be applied to enhance real-world systems:
>
> 1. **Genomic Sequence Alignment:** Seriation can be applied to reorder genomic sequences based on pairwise similarity metrics, such as sequence alignments or genetic distances. By grouping similar sequences together, seriation helps improve the alignment process.
>
> 2. **Recommendation Systems:** Seriation can be employed in recommendation systems to identify item rankings based on pairwise similarities between items (e.g., products or movies). A similarity matrix, derived from user-item interactions (e.g., ratings, clicks, or purchases), allows the algorithm to reorder items based on how similarly users interact with them. This approach helps uncover latent relationships between items, facilitating personalized recommendations even when the data is noisy. For example, if a product is liked by users who also like another product, the seriation algorithm can place these products closer together in the reordered similarity matrix, improving the overall accuracy of the recommendations.
>
> ---
>
> #  Response to Reviewer’s Comment on Numerical Experiments (Weaknesses 2):
>
> We agree with the reviewer and acknowledge that this point needs further clarification.  The primary contribution of our work is the development of an active seriation algorithm, which, to our knowledge, is the first of its kind. While we compare our algorithm to existing batch methods like Adaptive Sorting and Spectral Seriation in the (new) numerical experiments, it's important to note that these batch methods require the entire $n \times n$ matrix upfront. In contrast, our algorithm operates in an active setting where observations are sequentially collected, making the direct comparison somewhat limited.
>
> Regarding your comment on active learning or ranking algorithms, we would like to emphasize that there are no existing active seriation algorithms to compare against, as seriation and ranking are fundamentally different tasks. Active ranking algorithms, such as those proposed by Heckel et al. (2019), typically aim to rank items based on pairwise comparisons, often assigning intrinsic scores to individual items. However, seriation focuses on determining the relative ordering of items, which requires considering interdependencies between all pairwise similarities. Therefore, classical ranking methods based on pairwise comparisons cannot be directly applied to the seriation problem.

---

> > ### Comment · Reviewer_1ga3 · 2025-08-04
> > **Response to Authors**
> >
> > **Response to Reviewer's Comment on Noise Variance (Weaknesses 3 and 4, Question 1)**
> > Thank you for the clarification. While the algorithm does not explicitly require σ as input, the theoretical guarantees and appropriate choice of the total budget T still implicitly depend on knowledge of σ (as mentioned in the response too, T > function of σ). The first part of the response explains the dependency of T_0 on T, but T itself implicitly depends on σ.
> > In practice, selecting T without knowing σ may either violate the recovery guarantees or result in inefficient over-sampling. It would be helpful if the paper acknowledged this dependency more clearly and discussed whether adaptive estimation of σ or variance-aware sampling could mitigate this issue.
> >
> > **Response to Reviewer’s Comment on Dependence on Minimal Gap (Weakness 5, Question 2)**
> > Thank you for the thoughtful response. The proposed hybrid approach that modifies the Test subroutine to incorporate L_1 distances, seems promising, particularly as a means of improving robustness in low-gap or heterogeneous scenarios. However, the idea currently feels speculative without supporting analysis or empirical evidence. It remains unclear how much this modification improves over the original ASII algorithm, or how it compares to batch methods under realistic sampling constraints.
> > While the use of L_1 distances is intuitively appealing, it would be helpful to clarify how this modification interacts with the existing theoretical guarantees of ASII. For instance, does the use of approximate L_1 distances (via subsampling) still preserve the exponential decay in error? Or does it significantly change the algorithm’s convergence profile?
> > A clearer roadmap outlining how this hybrid mechanism would be implemented, analyzed, and evaluated would greatly strengthen the proposed approach.
> >
> > **Response to Reviewer’s Comment on Missing Data (Question 3)**
> > Thank you for addressing the possibility of handling missing data within the ASII framework. The suggestion to use subsampled L_1 distances to mitigate missing similarity entries is promising in principle. However, I have a few suggestions:
> > 1.	Your response mentions the need for the missing data proportion to be "small" but does not specify what range of sparsity is tolerable. It would be helpful to clarify what theoretical or empirical thresholds (if any) could guide this assumption.
> > 2.	While estimating L_1 distances from a few entries may reduce sampling cost, it might also introduce high variance into the Test decisions. I encourage the authors to more explicitly discuss how this trade-off could affect the algorithm's performance, especially under partial observations.
> >
> > **Response to Reviewer’s Comment on Real-World Applications (Weaknesses 1)**
> > Thank you for outlining these application areas. The proposed biological use cases and broader domains are promising, and I look forward to seeing empirical evaluations of ASII on these real-world datasets in future revisions.
> >
> > **Response to Reviewer’s Comment on Numerical Experiments (Weaknesses 2)**
> > Thank you for the clarification. I understand the challenge in identifying direct baselines for active seriation, and I appreciate the distinction drawn between seriation and traditional ranking tasks.

---

> > > ### Author Response · Authors · 2025-08-09
> > > **Response to Reviewer's Comment on Noise Variance**
> > >
> > > Due to the 5000-character limit for each answer, we respond to the 3 questions posed by the reviewer in 3 separate answers.
> > >
> > > # Response to Reviewer's Comment on Noise Variance
> > >
> > > We appreciate your feedback, and we'd like to clarify this further to avoid any confusion.
> > >
> > > ## I- Distinguishing Between Two Settings:
> > >
> > >  We believe the core of the concern lies in distinguishing between the following two settings:
> > > 1. **Type (1) Problem**: In this setting, the *total budget $T$ is fixed and known in advance*, and the goal is to *minimize the error probability* within this given budget. This is the setting addressed in our paper. Here, the total number of samples $T$ is predetermined, and we aim to make the most efficient use of the available resources.
> > >
> > > 2. **Type (2) Problem**: In contrast, in this setting, the *error probability is fixed and known in advance* (often referred to as `fixed confidence'), and the goal is to *minimize the total budget $T$* required to achieve the desired accuracy. In this scenario, the algorithm must adaptively query samples and stop once the error bound is met. The stopping rule often relies on adaptive estimation of parameters, including the noise level $\sigma$ and the gap $\Delta$.
> > >
> > > To clarify, our paper focuses on the **Type (1)** problem, where the total budget $T$ is fixed, and the goal is to minimize the error probability within that fixed budget.
> > >
> > > ## II- Clarification Regarding the Implicit Dependency on $\sigma$:
> > >
> > > We understand the concern about the implicit dependence of the algorithm's performance on $\sigma$. While our algorithm does not require explicit knowledge of $\sigma$, its performance is influenced by the signal-to-noise ratio (SNR) defined as $\text{SNR} := \frac{\Delta^2 T}{\sigma^2 n}$, which depends on $\sigma$. However, the key point is that the algorithm does **not** require prior knowledge of $\sigma$. Instead, it relies solely on the fixed budget $T$ available, and adapts to the noise level implicitly by determining the sample size $T_0$ based on $T$, as outlined in the rebuttal. This approach is analogous to how the algorithm adapts to $\Delta$.
> > >
> > > ## III- Discussion of the Extension to the Type (2) Problem:
> > >
> > > In the case of the Type (2) problem, where the error probability is fixed and the goal is to minimize the total budget, we agree that **adaptive estimation of $\sigma$** would be relevant. We also recognize that the Type (2) problem is of significant practical interest. While this is outside the scope of our current paper, we will mention it as an important open question in the revised manuscript.
> > >
> > > In the literature on classic Type (2) problems, $\sigma$ is often assumed to be known (e.g., best arm identification under fixed confidence [1] or active ranking under fixed confidence [2]). Even under this assumption, it is already challenging to adapt to the unknown signal $\Delta$. Moreover, in the context of our seriation problem, accurately estimating the variance is not straightforward, as we do not assume uniform variance across the matrix coefficients.
> > >
> > >
> > > ## Conclusion:
> > >
> > > We hope this distinction clarifies the scope of our work. We will revise the manuscript to emphasize this distinction and ensure that readers understand we are not addressing the **Type (2)** problem in this paper. Once again, thank you for your valuable feedback.
> > >
> > >
> > > [1] *An $\epsilon$-Best-Arm Identification Algorithm for Fixed-Confidence and Beyond*, by Jourdan, Degenne, Kaufmann, 2023
> > >
> > > [2] *Active Ranking of Experts Based on their Performances in Many Tasks*, by Saad, Verzelen, Carpentier, 2023

---

> > > ### Author Response · Authors · 2025-08-09
> > > **Response to Reviewer's Comment on Distance-Based Methods**
> > >
> > > ## Response to Reviewer's Comment on Distance-Based Methods
> > >
> > > We agree that incorporating $L_1$ distances, or other more advanced methods, into the Test subroutine could enhance robustness, particularly in low-gap and heterogeneous scenarios, as outlined in point 1 below. However, as discussed in point 2, several challenges remain in implementing this approach effectively.
> > >
> > > **Potential of Distance-Based Methods:**
> > >
> > >  1. **Flexibility of the Test Subroutine:** The ASII algorithm is  flexible, allowing the Test subroutine to be replaced with any function that returns correct orderings with high probability, including distance-based metrics (e.g. like $L\_1$?).
> > >  2. **Relaxing the $\Delta$-Separation Assumption:** Distance-based methods offer the potential to relax the $\Delta$-separation assumption. Rather than applying this assumption to every consecutive pair (as done in the definition of $\mathcal{M}_{\Delta}$ in the paper), it could be applied to a constant proportion of consecutive pairs (e.g., at least half of the items).
> > >
> > > **Challenges with Distance-Based Methods:**
> > >
> > > 1. **Toeplitz Assumption:** Distance-based methods typically assume a Toeplitz structure for the latent Robinson matrix $R$, which is a common assumption in batch seriation methods. (we did not adopt this assumption in our submitted work, as it is often considered restrictive.) Without the Toeplitz assumption, row distances may not correlate well with the latent orderings, thereby limiting the effectiveness of these methods.
> > > 2. **Bias of $L_1$-Estimation:** As noted by [Cai and Ma, 2022], their $L_1$-distance estimator is biased, particularly when $\Delta \leq \sigma$. The bias, which is of order $n\sigma$, can overwhelm the signal when $\Delta \leq \sigma$, making  $L_1$-based methods ineffective, especially in the challenging regime where $\Delta \leq \sigma$, which is the focus of our work.
> > > 3. **Limitations of $L_2$-Estimates:** The $L_2$-distance method, explored in [Issartel et al., 2024], provides more refined guarantees but results in suboptimal error bounds. For $\Delta \leq 1$, the error probability is expected to become $\exp\left(-\frac{\Delta^4 T}{\sigma^2 n \log(n)}\right)$, which worsens our current error rates by a factor of $\frac{\Delta^2}{\log(n)}$. Additionally, the $L_2$ estimator in [Issartel et al., 2024] is sensitive to the magnitude of matrix entries, requiring further assumptions (e.g., $\max_{i,j} |M_{ij}| \leq 1$).
> > >
> > >
> > >
> > >
> > > **Conclusion:** While distance-based methods have the potential to relax the $\Delta$-separation assumption (to some extent), they introduce new limitations, particularly in the important regime where $\Delta \leq \sigma$, which is the focus of our work. Our preliminary derivations of theoretical results seem either specific to restrictive regimes or lead to significantly suboptimal error rates, resulting in less clean outcomes. Additionally, our numerical simulations in the submitted version may have (unintentionally) favored distance-based methods due to poorly chosen scenarios. We plan to conduct further experiments with this hybrid approach and assess its performance in a broader range of settings.

---

> > > ### Author Response · Authors · 2025-08-09
> > > **Response to Reviewer’s Comment on Missing Data**
> > >
> > > ## Response to Reviewer’s Comment on Missing Data
> > >
> > > Thank you for your insightful comments. While missing data is not addressed in the current work, we offer the following observations:
> > > 1. **Handling Missing Data:**  In the current ASII algorithm, missing data prevents the Test subroutine from sampling missing pairwise similarities, resulting in partial orderings (instead of a total ordering $\pi$). The only slight modification required would be to ensure the algorithm operates only on the available (non-missing) pairwise similarities, but the overall analysis of the algorithm would remain the same.
> > > 2. **Impact of Missing Data on Distance-Based Methods:**  Distance-based methods are expected to be more robust to missing data since they rely on many pairwise similarities rather than on a single, specific pairwise similarity (as in our current Test subroutine discussed in point 1 above). As long as the remaining data provides a reasonable approximation of the distances (which holds under random missingness assumptions), the algorithm's performance should not be significantly impacted. Specifically, in the $\mathcal{M}\_{\Delta}$ setting considered in the paper, and under the Toeplitz structural assumption, we believe distance based algorithms could tolerate up to 50\% missing data without significant performance degradation. Indeed, in active problems, we do not rely on all matrix entries to estimate distances but instead sample a small subset of coordinates. Hence, missing data would not fundamentally alter the algorithm’s behavior; it would simply affect the sub-sampling of available entries. Assuming that missing data constitutes a small proportion (e.g., less than 20\%) and is *missing completely at random*, we expect no significant difference between sub-sampling with or without missing data (when assuming   the Toeplitz structure and $\mathcal{M}\_{\Delta}$ setting).  Therefore, the distance-based algorithm should still perform well. A detailed theoretical characterization of the impact of missing data on distance-based methods is beyond the scope of this paper, but we plan to discuss it further in the next version, and aim to explore it more thoroughly in future work.
> > > 3. **Variance and Subsampling:** Regarding the concern about variance introduced by sub-sampling, we agree that sub-sampling can increase the variance, e.g. of the $L_1$ distance estimates. However, since the current Test subroutine already involves sub-sampling (in some sense) and operates with a certain level of variance (as the test is correct with only constant probability), additional sub-sampling is unlikely to create new, significant issues. As a remark: if more reliable tests with smaller variance were desired, one could take $\log n$ additional samples, resulting in a $\log n$ factor increase in error rates.
> > >
> > > **Conclusion:** While missing data is not a primary focus of this work and we do not provide formal theoretical results for this case, we believe that small amounts of missing data, *under random missingness assumption*, are unlikely to pose new, fundamental obstacles to the ASII algorithm. This is particularly true when using distance-based methods instead of our Test subroutine. The reason is that active algorithms inherently rely on sub-sampling, meaning that missing completely at random data does not fundamentally change the algorithm’s behavior. We plan to include numerical simulations under missing data scenarios in the next version of the paper to illustrate the algorithm's performance in such conditions and provide a discussion of the results. While incorporating a rigorous theoretical analysis of missing data would be outside the scope of this work, we may add theoretical extensions that are not overly complex, as mentioned above. In any case, missing data and more robust methods (such as distance-based methods) are important areas for future research.

---

### Official Review · Reviewer_5WRt · 2025-06-30

**Clarity:** 3
**Significance:** 2
**Originality:** 2
**Rating:** 4
**Confidence:** 3

**Summary:**

This paper considers the problem of seriation, a noisy variant of sorting, from an active learning angle. The basic setting is as follows. There is an unknown ordering/ranking of $n$ items, where items that are closer to each other in the ranking are more similar than ones that are far apart. Specifically the similarities are encoded by a Robinson matrix, which is a symmetric matrix $A$ that is "monotone toward the diagonal". That is, any path in $A$ that starts at the entry $A_{n,1}$, ends at a diagonal entry, and only goes upward or to the right is monotone. Since the ordering is unknown, the algorithm only gets noisy access to some fixed permuted copy $M$ of such a Robinson matrix.

In the active seriation problem, the algorithm is active/adaptive, and in each of the $T$ rounds it selects a pair $i,j$ and obtains an independent draw from a $\sigma$-subgaussian random variable of mean $M_{ij}$. The goal is to retrieve the permutation (over the rows) exactly.
This paper assumes that there is a universal lower bound $\Delta$ on the similarities between elements and obtains tight results parameterized by this assumption. Specifically, the authors identify the SNR $\Delta^2 T / \sigma^2 n$ as the key quantity. They show that when this quantity is smaller than a constant times $\log n$, any algorithm fails with constant probability. On the other hand, they devise an algorithm whose failure probability is of order $e^{-O(SNR)}$. The algorithm is based on noisy binary search and includes an interesting ingredient of backtracking, that goes backwards in the search if it detects a possible violation. I did not go over the analysis (which appears only in the appendix).

**Questions:**

* There seems to be incompatibility between Theorem 3.1 and 3.2, specifically in the $\log n$ term. In Theorem 3.1 you claim that any algorithm where the SNR is less than $\log n$ fails with constant failure probability, whereas in Theorem 3.2 you obtain an algorithm that achieves constant failure prob already when the SNR is a constant. Please explain how these settings differ.

* Why does the backtracking work? I was curious about more details about the theoretical analysis (and in contrast am not that interested in the experiments). Backtracking is not that common in noisy binary search approaches, and introducing such a technique could be a valuable contribution in your paper, which you don't emphasize enough.

* How would your results change if you only asked for a ranking that is approximately monotone?

**Ethical Concerns:**

["NO or VERY MINOR ethics concerns only"]

**Final Justification:**

I still view it as a borderline paper, but given the positive reception by others, flipped it from a weak reject to a weak accept score. In essence, there are interesting and novel theoretical contribution (the backtracking algorithm), and on the other hand the bottom line results (and the dependence in the parameters) are perhaps a slight weakness.

**Limitations:**

yes

**Quality:**

2

**Strengths And Weaknesses:**

+ The studied problem is interesting, and natural for an active approach.

- Technical novelty is not very high. Noisy binary search is probably the first technique that one would try in such a setting, and the analysis does not seem to carry with it very deep lessons or surprises.

- The dependence in the minimum similarity $\Delta$ seems rather limiting. For example, it could be the case that we rank a large set of elements where some pairs of them are near-identicals, and we wish to get a good ranking at least for those elements that are far enough. It would be better if you can extend your results to this regime (which I believe shouldn't be too hard).

-+ There are some issues in terms of writing, although overall it is decent. One example of an issue is my first question to the authors (which claims to Theorems 3.1 and 3.2 are not completely compatible with each other). Also, given that the technical analysis is claimed to be the main contribution in this work, at least a snippet of it should appear in the main body.

---

> ### Author Rebuttal · Authors · 2025-07-31
>
> Thank you for your thoughtful feedback, and for taking the time to review our work. We have addressed your concerns by rewriting the entire paper, especially the algorithm section, to provide clearer intuition on how and why the algorithm works. We hope the revised version of the paper will better explain the key contributions.
>
> ---
>
> # Response to Reviewer's Comment on Incompatibility between Theorems (1st Question)
>
> We understand the source of confusion and would like to clarify that the regimes in Theorem 3.1 and Theorem 3.2 are complementary, and *not* incompatible. Specifically:
>
>  1. In **Theorem 3.1**, we consider the *impossibility regime*, where the signal-to-noise ratio (SNR) satisfies
>     $\frac{\Delta^2 T}{\sigma^2 n} \lesssim \log n \enspace.$
>     In this setting, we show that no algorithm can recover the correct ordering with vanishing error probability.
>
> 2. In contrast, **Theorem 3.2(a)** addresses the *recovery regime*, where
>     $\frac{\Delta^2 T}{\sigma^2 n} \gtrsim \log n \enspace.$
>     Under this condition, we provide performance guarantees for our ASII algorithm, showing that the error probability decays exponentially fast with the SNR.
>
>
> We acknowledge that **Theorem 3.2(b)**, which provides a lower bound on the error rate for any algorithm, could be confusing, as it overlaps with the regime in **Theorem 3.1** by applying when $\frac{\Delta^2 T}{\sigma^2 n} \gtrsim 1$.
>
> However, this does not contradict **Theorem 3.1**. Instead, it complements it: **Theorem 3.2(b)** shows that in the recovery regime 2 (above), no algorithm can achieve error decay faster than exponential in the SNR, while **Theorem 3.1** asserts that in the complementary regime  1 (above), the error probability remains lower bounded by a constant.
>
> To enhance clarity, we will revise the statements of Theorems 3.1 and 3.2 to explicitly state the respective conditions of the regimes 1 and 2 (above) as *numbered equations* within the theorems themselves. We also recognize that stating **Theorem 3.2(b)** under the more general condition
> $\frac{\Delta^2 T}{\sigma^2 n} \gtrsim 1$,
> rather than the recovery regime 2 (above), may have caused confusion without adding benefit, and we will revise accordingly.
>
> ---
>
> # Response to Reviewer's Comment on the Backtracking Mechanism (2nd Question)
>
> Thank you for your insightful comment. We are glad that you find the backtracking approach intriguing, and we are happy to provide further clarification on its role in our **Binary-\&-Backtracking-Search (BBS)** subroutine.
>
> In noisy binary search, incorrect decisions at any step can propagate and lead to compounding errors, ultimately resulting in incorrect item placement. Traditional methods mitigate this by increasing the number of samples per test to reduce error probability, but this approach can be sample-inefficient. In contrast, our method uses a fixed, small number of samples per test, ensuring constant accuracy (e.g., a correct decision probability of at least $3/4$). This means there is a small probability of error at each decision point.
>
> Backtracking serves as a corrective mechanism to address these errors. The algorithm maintains a history of explored intervals and performs sanity checks at each step. If an inconsistency is detected, the algorithm backtracks to a previous interval and resumes the search. This prevents a single mistake from permanently misguiding the process.
>
> The combination of low-sample testing and backtracking ensures that, despite occasional errors, the overall number of correct decisions dominates the incorrect ones over the search path. As a result, the algorithm achieves a high probability of correct insertion while maintaining near-optimal sample efficiency.
>
> We will expand this discussion of this mechanism in the revised manuscript to more clearly explain how it works and its importance in the procedure.
>
> Additional clarification on the backtracking mechanism::
>
>
> 1. Using backtracking in our binary search algorithm allows us to tolerate a certain number of mistakes, as long as enough good decisions are made to offset them. Specifically, if we perform enough steps, even a constant proportion of mistakes can be corrected by backtracking, ensuring that the final result is correct. The key point here is that we can afford some mistakes, as long as they are balanced by correct decisions that guide us to the correct point of insertion.
> 2. In contrast, if we remove backtracking, the algorithm becomes more sensitive to mistakes at each stage. Without the corrective mechanism, an error at any step could result in an incorrect placement. This would make the algorithm less reliable than the version with backtracking.
>
>
> ---
>
> # Response to Reviewer's Comment on Extension to Approximate Seriation (3rd Question)
>
> Thank you for this insightful comment. We agree that the dependence on the minimum similarity can be limiting in some cases. While our current analysis focuses on exact monotonicity, we acknowledge that extending the results to approximately monotone rankings --only reordering elements that are sufficiently dissimilar-- is an important direction.
>
> Unfortunately, our current binary search method struggles in this case. When items are too close to each other (less than $\Delta$ apart), there are no longer guarantees of a "correct answer" in the tests performed along the search, and the binary and backtracking search can become unpredictable, leading to instability in the ordering.
>
> We see two potential approaches to address this issue:
> 1. **Simple, Sub-Optimal Solution:** A straightforward method would be to remove backtracking and sample $\ln n$ times more per test. This would ensure high-probability correctness at each decision, resulting in a simple algorithm that works for items sufficiently far apart. While this solution is less sample-efficient (increasing the number of samples by an extra logarithmic factor), it provides a reasonable approach for handling very similar items. For example, if the items are partitioned into groups $G_1, \dots, G_K$, with items in the same group being identical, the algorithm would correctly reorder items between groups that are at least $\Delta$ apart (i.e., $R_{i,j} - R_{i,k} \geq \Delta$ whenever $j$ and $k$ are from different groups, for all $i$).
> 2. **Difficult, Optimal Solution:** A more challenging extension would involve modifying the binary and backtracking search. Instead of placing an item at the last visited interval, we would track all small intervals visited and place the item in the "median" of those intervals. This would allow the algorithm to better handle identical or near-identical items, but it would require significant theoretical development of our current proofs. While we cannot promise this will appear in the camera-ready version of the paper, we are committed to exploring this direction and sincerely hope to obtain concrete results before the final submission.

---

> > ### Comment · Reviewer_5WRt · 2025-08-07
> > **Thanks**
> >
> > Thanks to the others for their detailed response to my comments/concerns. This is an interesting work, and although I am still not completely satisfied by the results given the dependence in the minimum distance, I am fine with acceptance especially due to the introduction of the backtracking technique. To align with the other reviewers (and since my view of the paper is as a borderline one), I increase my score to borderline accept.

---

> > > ### Author Response · Authors · 2025-08-09
> > > **Response to Reviewer’s Final Comment:**
> > >
> > > Thank you for your thoughtful feedback and for increasing your score.
> > >
> > > Regarding your concern about the dependence on the minimal gap $\Delta$, we fully acknowledge the limitations this introduces. As discussed earlier in our response, we plan to include the median trick (referred to as the "Difficult, Optimal Solution" in our previous reply) to address this issue, and we believe it will improve the algorithm's robustness when handling very similar items, and allow to output approximate rankings with guarantees on distant items. We are working on proving this extension, and hope to include them in the final version of the paper.
> > >
> > > Additionally, we refer to our answers to "Reviewer 1ga3" for potential solutions that further address these concerns, see the answers "Dependence on Minimal Gap $\Delta$" and "Distance-Based Methods")
> > >
> > > Once again, thank you for your constructive feedback!

---

### Official Review · Reviewer_HzA5 · 2025-07-03

**Clarity:** 3
**Significance:** 3
**Originality:** 3
**Rating:** 5
**Confidence:** 4

**Summary:**

This paper studies the active seriation problem: i) there are n items with a hidden true ranking; ii) the learner can query two items each time, and each query returns a noisy sample of the similarity of them; iii) If in the true ranking, two items are closer, then the similarities are closer. The learner wants to recover the full ranking with high probability by actively selecting pairs to query, and tries to use the least amount of queries.

The authors gives the lower bound and upper bound on the sample complexity (O(n\log(n)), and proposes a new algorithm achieving the optimal sample complexity.

**Questions:**

Maybe give some concrete examples to explain the real-world use case of this paper's setting?

**Ethical Concerns:**

["NO or VERY MINOR ethics concerns only"]

**Limitations:**

No negative social impacts found

**Paper Formatting Concerns:**

No formatting issues found

**Quality:**

3

**Strengths And Weaknesses:**

Strengths
-The problem studies in this paper is interesting and fundamental in my opinion, and the authors give clear and optimal results on the lower and upper bounds. The settings are more general than previous works. Overall, the results and novelty of this paper are pretty good. I find no major flaws

Weakness
-For readers not quite familiar with this problem, maybe give more concrete real-world examples about this paper's setting.

---

> ### Author Rebuttal · Authors · 2025-07-31
>
> # Response to Reviewer’s Comment on Real-World Applications (Question)
>
>  We appreciate the reviewer's suggestion to highlight real-world applications of our method. To strengthen the significance of our work, we refer to the recent study by Cai and Ma (2022), which applies seriation reordering techniques to real-world biological data. In their study, two datasets were analyzed to infer the temporal ordering of cells in single-cell RNA sequencing experiments. We plan to apply our active seriation algorithm to these same datasets and present our results in the updated version of our paper, comparing the performance with that of Cai and Ma’s method.
>
> 1. **Real Dataset 1:** The first dataset consists of 372 primary human skeletal muscle myoblasts undergoing differentiation at 24-hour intervals. The goal was to infer the temporal order of cells during differentiation. By constructing similarity matrices and applying a seriation algorithm,  Cai and Ma (2022) demonstrated the good performance of their algorithm in recovering the true temporal order.
>
> 2. **Real Dataset 2:** The second dataset involves 149 human primordial germ cells from various ages ranging from 4 to 19 weeks. A similar methodology was used to infer the temporal order of these cells.
>
>
> These real-world applications highlight the practical utility of seriation algorithms in analyzing complex, noisy, and high-dimensional biological data. The ability to recover latent temporal or sequential structures from such datasets is a crucial problem in computational biology. Our method offers a computationally efficient and statistically robust solution, particularly when data is  noisy. Moreover, one of the key advantages of our active seriation algorithm is its ability to achieve accurate results with fewer similarity measurements, which can be of practical interest in scenarios where obtaining large amounts of pairwise comparisons may be costly or time-consuming.
>
> Additionally, we highlight several other domains where our seriation method can be applied to enhance real-world systems:
>
>
>  1. **Genomic Sequence Alignment:** Seriation can be applied to reorder genomic sequences based on pairwise similarity metrics, such as sequence alignments or genetic distances. By grouping similar sequences together, seriation helps improve the alignment process, especially in noisy or incomplete datasets, making it more efficient and robust for large-scale genomic comparisons.
>
>
> 2. **Recommendation Systems:** Seriation can be employed in recommendation systems to identify item rankings based on pairwise similarities between items (e.g., products or movies). A similarity matrix, derived from user-item interactions (e.g., ratings, clicks, or purchases), allows the algorithm to reorder items based on how similarly users interact with them. This approach helps uncover latent relationships between items, facilitating personalized recommendations even when the data is noisy. For example, if a product is liked by users who also like another product, the seriation algorithm can place these products closer together in the reordered similarity matrix, improving the overall accuracy of the recommendations.

---

### Official Review · Reviewer_FHVh · 2025-07-05

**Clarity:** 2
**Significance:** 3
**Originality:** 3
**Rating:** 4
**Confidence:** 3

**Summary:**

This paper explores the active seriation problems. On the one hand, it demonstrates the impossibility of solving active seriation problems when the SNR value is smaller than $\log n$ orderwise, where $n$ denotes the number of items, indicating the difficulty of this problem. On the other hand, when SNR exceeds $\log n$, this paper proposes Active Seriation by Iterative Insertion (ASII) algorithm, which matches the lower bound of the error metric. In addition, some numerical results also validate the effectiveness of the proposed algorithm.

**Questions:**

Please try to answer the following questions:

1. (Main) I believe the active seriation problem is closely related to MAB problems based on my experience: The sub-Gaussian assumption, noisy observations, and the form of the upper and lower bounds are quite similar to those in the MAB literature. Although the authors also briefly mentioned this point in Line 55, the subsequent discussion is missing. My questions regarding this connection are twofold: (1) Theoretically, can the authors provide more discussions about their similarities and differences? I wonder if this problem can even be formulated as an MAB problem? If not, what is the key distinction? (2) Experimentally, can some popular MAB algorithms such as UCB, TS, be adopted as baselines here? A comparison with these methods might help demonstrate the effectiveness of the proposed algorithm.

2. For the experiments: (1) The Fig. 3 cannot reflect how parameters $\sigma, n, T$ impact the performance of ASII, which are key components in the theorem. (2) In scenario 3, AS algorithm dramatically outperforms the others, and achieves small errors ($\le 0.2$) when $\beta$ is small. Does this violate Theorem 3.1? (3) Can the authors provide the error bar to show the robustness of the proposed algorithm?

3. Can the authors give some real-world applications to strengthen the significance of this paper?

**Ethical Concerns:**

["NO or VERY MINOR ethics concerns only"]

**Limitations:**

Please focus on Questions. I think there are no additional limitations.

**Paper Formatting Concerns:**

No.

**Quality:**

3

**Strengths And Weaknesses:**

Strengths:

1. This discussion of SNR $\sim \log n$ and the results on lower bound make this paper complete. When SNR is small, the difficulty of active seriation problems is well clarified, delineating the boundaries of problem solvability. In addition, the proposed lower bound strengthens the theoretical foundation of ASII algorithm.

2. The proposed ASII algorithm is both elegant and effective. Although the underlying idea is quite simple, the analysis is concrete, the detailed design, such as the improved binary search, is dedicated, the theoretical results are strong (matching the lower bound orderwisely), and the numerical results are also convincing.

Weaknesses:

1. The presentation is somewhat unclear. The structure of the paper is confusing, for example, the theoretical analysis of ASII algorithm appears before the statement of the algorithm itself. Moreover, many typos lead to confusions and reduce readability, e.g., (1) Should $\Delta(M)$ be $\Delta_M$ in Line 126? (2) How many scenarios are considered in numerical experiments, as the caption of Fig. 2 seems inconsistent with the discussion? (3) There are some minor grammar mistakes, such as “exists” in Line 274.

---

> ### Author Rebuttal · Authors · 2025-07-31
>
> Thank you for your valuable feedback. We apologize for the issues with the presentation and organization of the manuscript. In response, we have rewritten several sections of the paper. We hope these revisions have greatly improved its clarity and readability.
>
> ---
>
>
> #  Response to Reviewer's Comment on MAB (Question 1):
>
> While Multi-Armed Bandit (MAB) algorithms focus on selecting arms based on independent observations to maximize cumulative rewards (or identify the best arm in best-arm identification), the active seriation problem has a fundamentally different goal. In MAB, each arm provides an independent reward, and the objective is to identify the arm with the highest expected reward over time. Classical algorithms like UCB or TS are designed to concentrate sampling on the best arms (those with the highest expected rewards).
>
> In contrast, active seriation involves interdependent observations: the similarity between two items reflects their relative positions in an underlying ordering. Querying one pair of items can thus influence how we interpret the similarity of other pairs, because the relationships between all pairs must be consistent with the underlying ordering. This interdependence makes the problem more complex than simply optimizing individual rewards, as the goal is to recover the entire ordering (permutation) of the items. Unlike MAB, which optimizes individual rewards, seriation's challenge is to determine this global ordering from pairwise measurements, making it a fundamentally different problem.
>
> Thus, while both problems are sequential and involve repeated querying, active seriation aims to recover a permutation of items, a combinatorial task, which differs from maximizing rewards or identifying the best arm in MAB. This is why classical MAB algorithms like UCB or TS, which assume independent rewards, cannot be directly applied to seriation.
>
> A more closely related problem in the literature is active ranking [Heckel et al., 2019], where the goal is to rank items based on their expected rewards using pairwise comparisons. However, ranking problems often assign intrinsic scores to individual items, whereas seriation does not. Instead, seriation focuses on recovering the relative ordering of items, which requires considering interdependencies between all pairwise similarities. For this reason, classical ranking methods based on pairwise comparisons cannot be directly applied to seriation either.
>
> ---
>
> #  Response to Reviewer's Comment on Experiments (Question 2):
>
> **Clarification:** The primary contribution of our work is the development of an active seriation algorithm, which, to our knowledge, is the first of its kind. In the numerical experiments, we compare our algorithm to existing batch methods, which require the full $n \times n$ matrix upfront. While we place our algorithm in a batch setting for comparison, it's important to note that this setting is not directly relevant to our active approach, where observations are collected sequentially. For instance, if only $T$ observations are available with $T < n^2$, batch algorithms cannot function, whereas our algorithm can still proceed and yield meaningful results.
>
>
>
> **Improvement of the Experimental Section:** We  apologize for the shortcomings in the experimental section of our original submission.  Since the submission, we have completely rewritten and significantly improved the experimental section, incorporating new simulations that  highlight the strengths of our ASII algorithm.
>
> In particular, we focused on addressing one of the major limitations of previous work: the reliance on the Toeplitz assumption for certain algorithms, such as the AS algorithm (Cai and Ma, 2022). Our experiments now demonstrate that the performance of ASII is independent of the Toeplitz structure, making it applicable to a much broader class of matrices. In contrast, the AS algorithm performs poorly on heterogeneous matrices, due to its reliance on properties specific to Toeplitz matrices.
>
> We also introduced the famous spectral seriation algorithm as a new competitor in our experiments. However, as noted in prior work (e.g., Cai and Ma, 2022), the spectral algorithm does not have strong theoretical guarantees for general matrix classes, and this is reflected in its performance outside the Toeplitz setting. We believe that these new simulations and insights provide a clearer, more rigorous comparison between algorithms and better illustrate the advantages of our proposed method.
>
>
> ## 1) Impact of Parameters $\sigma, n, T$ on Performance:
>
> We appreciate the reviewer's suggestion to explore the impact of the parameters $\sigma$, $n$, and $T$ more explicitly in our experiments. In the revised version of the paper, we have conducted additional experiments to better illustrate how these parameters influence the performance of the ASII algorithm.
>
>
>
> ## 2) Performance of AS Algorithm in Scenario 3 and Violation of Theorem 3.1?
>
> Regarding the observed performance of the AS algorithm in Scenario 3, where it dramatically outperforms the others and achieves small errors when $\beta$ is small, this scenario is extremely favorable for the AS algorithm and unfavorable for the ASII procedure. The performance of the AS algorithm depends on a global signal in the matrix (specifically, variations in the $L_1$-distance between rows), while the ASII procedure relies on a local signal in the matrix (the minimal gap $\Delta$ as defined in the paper).
>
> In this scenario, where $\beta = \Delta$, the local signal diminishes as $\Delta \rightarrow 0$, while the global signal remains large, regardless of $\Delta$ values. This explains why the AS algorithm outperforms ASII in this case.
>
> We acknowledge that this numeric example may have been misleading and not well-chosen. We have revised our numerical section to clarify this point. Importantly, the above experiment does not contradict the lower bound of Theorem 3.1. Recall the notation $\mathcal{M}\_{\Delta}$ for the class of similarity matrices with minimal gap $\Delta$ (and $\mathcal{R}\_{\Delta}$, which was incorrectly used in the original manuscript as a typo). The impossibility result (Theorem 3.1) represents a worst-case analysis over the class $\mathcal{M}\_{\Delta}$: the lower bound is proved for a specific instance in the class $\mathcal{M}\_{\Delta}$. For example, when $\Delta = 0$, the instance is the null matrix $M = \mathbf{0}_{n \times n}$, which indeed belongs to $\mathcal{M}_0$. However, the matrix considered in Scenario 3 of our experiment is not the null matrix when $\beta = \Delta = 0$, and therefore, Theorem~3.1 does not apply to this matrix. Hence, there is no contradiction between Theorem 3.1 and our original experiments.
>
>
>
> ##  3) Error Bar for Robustness:
>
> We acknowledge the importance of demonstrating the robustness of the proposed algorithm through error bars. In the revised paper, we will include error bars in our experimental results.
>
>
> ---
>
> #   Response to Reviewer's Comment on Real-World Applications (Question 3)
>
> We appreciate the reviewer's suggestion to highlight real-world applications of our method. To strengthen the significance of our work, we refer to the recent study by Cai and Ma (2022), which applies seriation reordering techniques to real-world biological data. In their study, two datasets were analyzed to infer the temporal ordering of cells in single-cell RNA sequencing experiments. We plan to apply our active seriation algorithm to these same datasets and present our results in the updated version of our paper, comparing the performance with that of Cai and Ma’s method.
>
> 1. **Real Dataset 1:** It consists of 372 primary human skeletal muscle myoblasts undergoing differentiation at 24-hour intervals. The goal was to infer the temporal order of cells during differentiation. By constructing similarity matrices and applying a seriation algorithm,  Cai and Ma (2022) demonstrated the good performance of their algorithm in recovering the true temporal order.
>
> 2. **Real Dataset 2:** It involves 149 human primordial germ cells from various ages ranging from 4 to 19 weeks. A similar methodology was used to infer the temporal order of these cells.
>
>
> These real-world applications highlight the practical utility of seriation algorithms in analyzing complex, noisy, and high-dimensional biological data~[1]. The ability to recover latent temporal or sequential structures from such datasets is a crucial problem in computational biology. Our method offers a computationally efficient and statistically robust solution.
> Moreover, one of the key advantages of our active seriation algorithm is its ability to achieve accurate results with fewer similarity measurements, which can be of practical interest in scenarios where obtaining large amounts of pairwise comparisons may be costly or time-consuming.
>
> Additionally, we highlight several other domains where our seriation method can be applied to enhance real-world systems:
>
>
>  1. **Genomic Sequence Alignment:** Seriation can be applied to reorder genomic sequences based on pairwise similarity metrics, such as sequence alignments or genetic distances.
>
> 2. **Recommendation Systems:** Seriation can be employed in recommendation systems to identify item rankings based on pairwise similarities between items (e.g., products or movies). For example, if a product is liked by users who also like another product, the seriation algorithm can place these products closer together in the reordered similarity matrix, improving the overall accuracy of the recommendations.
>
>
> [1] scPrisma infers, filters and enhances
> topological signals in single-cell data using
> spectral template matching, Jonathan et al., 2023, Nature Biotechnology

---

> > ### Comment · Reviewer_FHVh · 2025-08-09
> >
> > The authors have addressed my questions satisfactorily. I will maintain my original positive assessment.

---

### Decision · Program_Chairs · 2025-09-17

**Decision:**

Accept (poster)

**Comment:**

This paper considers a novel problem of active, sequential hypothesis testing: how to query pairs of items, observe (noisily) a signal about (the magnitude of) their relative similarity, and use this information to rank-order them so that their relative positions correlate with their true underlying similarities. It carries out a comprehensive study in the sense of (a) providing an information-theoretic limit for low-error recovery by any procedure and (b) proposing an efficient algorithm, based on a backtracking approach and a generalization of insertion sorting, that is proven to be optimal when the signal in the problem is above the information-theoretic SNR limit.

The paper's initial reviews asked for clarifications on several points, including connections with the multi-armed bandit literature, real-world motivating applications for the active seriation model, the relationship between the achievability and converse results of the paper, and the possibility of deriving schemes under only approximate monotonocity and highly skewed similarity structures. The authors responded with detailed clarifications on many of these points, which was noted positively by the reviewers.

Overall, barring the concerns about practical motivating situations for this rather general ordering model (which seems to be weaker than explicit win-loss feedback), the reviewers unanimously lean positive about the paper's contributions and its potential impact in introducing a new decision-making model. In view of the consensus and the strengths of the paper, I am happy to recommend acceptance.